# The Influence of Sex Steroid Hormone Fluctuations on Capsaicin-Induced Pain and TRPV1 Expression

**DOI:** 10.3390/ijms25158040

**Published:** 2024-07-24

**Authors:** Edgardo Mota-Carrillo, Rebeca Juárez-Contreras, Ricardo González-Ramírez, Enoch Luis, Sara Luz Morales-Lázaro

**Affiliations:** 1División de Neurociencias, Instituto de Fisiología Celular, Universidad Nacional Autónoma de México, Ciudad de México 04510, Mexico; emota@ifc.unam.mx (E.M.-C.); rjuarez@ifc.unam.mx (R.J.-C.); 2Programa de Doctorado en Ciencias Biomédicas, Unidad de Posgrado, Universidad Nacional Autónoma de México, Ciudad de México 04510, Mexico; 3Programa de Posgrado en Ciencias Biológicas, Unidad de Posgrado, Universidad Nacional Autónoma de México, Ciudad de México 04510, Mexico; 4Departamento de Biología Molecular e Histocompatibilidad, Hospital General “Dr. Manuel Gea González”, Ciudad de México 14080, Mexico; ricardo.gonzalezr@salud.gob.mx; 5Centro de Investigación sobre el Envejecimiento, CINVESTAV, Ciudad de México 14390, Mexico; 6Investigador por México—Instituto de Fisiología Celular, Universidad Nacional Autónoma de México, Ciudad de México 04510, Mexico; enoch@ifc.unam.mx; 7Laboratorio Nacional de Canalopatías, Instituto de Fisiología Celular, Universidad Nacional Autónoma de México, Ciudad de México 04510, Mexico

**Keywords:** pain, estrogens, TRPV1, hormones, diestrus, proestrus

## Abstract

Sexual dimorphism among mammals includes variations in the pain threshold. These differences are influenced by hormonal fluctuations in females during the estrous and menstrual cycles of rodents and humans, respectively. These physiological conditions display various phases, including proestrus and diestrus in rodents and follicular and luteal phases in humans, distinctly characterized by varying estrogen levels. In this study, we evaluated the capsaicin responses in male and female mice at different estrous cycle phases, using two murine acute pain models. Our findings indicate that the capsaicin-induced pain threshold was lower in the proestrus phase than in the other three phases in both pain assays. We also found that male mice exhibited a higher pain threshold than females in the proestrus phase, although it was similar to females in the other cycle phases. We also assessed the mRNA and protein levels of TRPV1 in the dorsal root and trigeminal ganglia of mice. Our results showed higher TRPV1 protein levels during proestrus compared to diestrus and male mice. Unexpectedly, we observed that the diestrus phase was associated with higher TRPV1 mRNA levels than those in both proestrus and male mice. These results underscore the hormonal influence on TRPV1 expression regulation and highlight the role of sex steroids in capsaicin-induced pain.

## 1. Introduction

Neuronal activity triggered by harmful stimuli produces a subjective response known as pain [1]. These dangerous signals are detected through specialized peripheral neurons, called nociceptors, which express specific transmembrane protein complexes that transform noxious stimuli into electrochemical signals [2,3].

Among these proteins, Transient Receptor Potential Vanilloid 1 (TRPV1) is a non-selective cation channel essential for transducing injurious messages from the compounds released during inflammation and tissue damage [4,5,6].

TRPV1 is abundantly expressed in nociceptors from the dorsal root and trigeminal ganglia (DRGs and TGs, respectively). It acts as a molecular sensor for several noxious stimuli, such as temperatures ≥43 °C, changes in pH (acid and alkaline), natural chemicals (such as capsaicin and resiniferatoxin), and endogenously produced compounds, such as lysophosphatidic acid, anandamide, and products derived from arachidonic acid, among others [6,7,8]. 

Several studies have reported the overexpression of TRPV1 in certain diseases such as rheumatoid arthritis, neuropathic pain, bone cancer, fibromyalgia, migraine, and irritable bowel syndrome [9,10,11,12]. Some of these painful conditions are more frequent in women than in men, indicating that sex is a biological variable that influences the pain produced through TRPV1 activation [13,14,15]. 

In the last 20 years, there has been growing interest in understanding the influences of sex differences on pain perception [16]. The differences in pain thresholds between sexes are strongly influenced by sexual steroid hormones, which exert molecular and cellular effects on the physiology of nociceptors, contributing to sexual dimorphism in which females typically exhibit a lower pain threshold than males [17]. Recognizing this, the International Association for the Study of Pain (IASP) declared 2007 as the Global Year Against Pain in Women [18], publishing a collection of reports evidencing that women are more susceptible to certain painful conditions than men. These conditions include fibromyalgia, irritable bowel syndrome, temporomandibular disorder, rheumatoid arthritis, osteoarthritis, and migraines (IASP, 2007), all of which are associated with TRPV1 upregulation. 

The role of TRPV1 in the sexual dimorphism of pain has been demonstrated in several experimental models, ranging from animals to humans. For instance, intradermal capsaicin application in the foreheads of male and female subjects revealed that females experienced greater pain than males [19]. Similarly, experiments performed in rodents have shown that females exhibit lower mechanical, thermal, and capsaicin pain thresholds than males. Additionally, some reports indicate that rodents’ estrous cycle phases affect the capsaicin-induced nocifensive response [20,21], suggesting that steroid hormonal fluctuations are also a biological variable influencing pain perception. For example, female rodents exhibit low pain thresholds during proestrus, an estrous cycle phase with high estrogen levels [21]. Furthermore, ovariectomized rats display a decreased capsaicin pain response; this effect is reversed by estrogen replacement [20], suggesting that estrogen has a positive role in the pain associated with TRPV1 activation.

These differences point to TRPV1 expression regulation through the genomic classic estrogen effects, since DRG primary cultures treated with 17β-estradiol (E2) display an upregulation of TRPV1 mRNA levels [22]. The estrogen genomic actions are through the estrogen receptors alpha and beta (ERα and ERβ), transcription factors that bind to specific DNA sequences (known as Estrogen Response Element, ERE), regulating the transcription of different gene targets [23]. Thus, estrogen actions on TRPV1 expression regulation could be through this classical pathway; furthermore, some ERE sequences identified in the TRPV1 gene regulatory region support this idea [24].

Besides the evidence, no reports show the influence of sexual steroid fluctuations on TRPV1 gene expression in the DRG or TG isolated from female mice in different estrous cycle phases. This study, which evaluated the capsaicin-induced nocifensive response using two acute pain models in male and female mice at different estrous cycle phases and assessed TRVP1 gene expression, has significant implications. We observed sexual dimorphism in the capsaicin-evoked nocifensive response in both pain models. Interestingly, this difference depended on the estrous cycle phase of the females; specifically, the proestrus phase exhibited a lower capsaicin-evoked pain threshold than those of males and females in the other phases. TRPV1 protein levels in the DRG and TG were lower in the diestrus phase than in the proestrus phase, correlating with the results obtained in behavioral experiments, which showed less protein and less pain. Furthermore, calcium imaging recordings performed in primary cultures of TG neurons confirmed that cells treated overnight with a high E2 concentration, simulating the proestrus condition, exhibited a faster activation response after capsaicin application compared to neurons cultured in control and low E2 concentration conditions (simulating diestrus). Unexpectedly, the TRPV1 mRNA levels were higher during diestrus than during proestrus and in male mice, suggesting that TRPV1 gene expression during sex steroid fluctuations is regulated at both the transcriptional and post-transcriptional levels. Finally, our data demonstrated a correlation between ERα and TRPV1 protein expressions in two phases of the estrous cycle: during the proestrus phase, we found higher TRPV1 and ERα protein levels than in the diestrus phase, strengthening the putative role of ERα as a positive regulator of TRPV1 expression. These findings have significant implications for understanding the regulation of TRPV1 gene expression and its impact on pain perception.

## 2. Results

### 2.1. Effects of Sexual Steroid Hormone Fluctuations on Capsaicin-Induced Pain in Mice 

We evaluated the effects of sex steroid hormones on pain induced by TRPV1 activation through intradermal capsaicin injection in the paws of male and female mice at different phases of the estrous cycle. The capsaicin-evoked nocifensive response was lower in male mice than in females in proestrus, although similar to estrus, metestrus, and diestrus (males: 51 ± 3 s; proestrus: 79 ± 5 s; estrus: 51 ± 4 s; metestrus: 45 ± 3 s; and diestrus: 47 ± 3 s) (Figure 1A). We observed that female mice in proestrus exhibited a lower capsaicin-evoked pain threshold compared to estrus, metestrus, and diestrus phases, Figure 1A. Mice injected with the vehicle solution showed a reduced nocifensive response compared to mice injected with capsaicin (males: 18 ± 1 s; proestrus: 27 ± 2 s; estrus: 19 ± 2 s; metestrus: 23 ± 4 s; and diestrus: 21 ± 1 s). 

Capsaicin-induced pain was also evaluated using the pain cheek model in male and female mice during the estrous cycle. The nocifensive response, displayed as wiping bouts, was higher in females in proestrus than in male and female mice in other phases of the estrous cycle (number of wiping bouts: males: 59 ± 7; proestrus: 113 ± 9; estrus: 62 ± 8; metestrus: 26 ± 9; and diestrus: 46 ± 7). The injection of vehicle solution evoked few wiping bouts in male and female mice (males: 21 ± 5; proestrus: 23 ± 2; estrus: 23 ± 8; metestrus: 14 ± 4; and diestrus: 22 ± 5) (Figure 1B). These data revealed that sex steroid hormone fluctuations in female mice impact capsaicin-induced pain behavior.

### 2.2. TRPV1 Gene Expression Is Differentially Regulated during Proestrus and Diestrus 

The differences in the pain threshold found in female mice during proestrus and diestrus could be associated with changes in TRPV1 gene expression. To investigate this, we evaluated TRPV1 mRNA and protein expression in the DRGs isolated from female mice in proestrus and diestrus and compared it to TRPV1 expression in male mice. The results showed that TRPV1 relative mRNA levels were similar in the DRGs of both sexes, (male: 1 ± 0.13, proestrus: 0.88 ± 0.24, and diestrus: 1.07 ± 0.01) (Figure 2A). We also observed a tendency toward higher TRPV1 mRNA expression in the diestrus phase than in the proestrus phase, although the differences were not statistically significant. 

TRPV1 protein levels were evaluated in the proestrus and diestrus phases in the DRGs of female and male mice. We found that TRPV1 protein expression was higher in the proestrus phase than in the diestrus phase. Similar TRPV1 protein levels were observed in the DRGs isolated from male compared to female mice in the proestrus phase, (male: 0.88 ± 0.09, proestrus: 0.99 ± 0.13, and diestrus: 0.7 ± 0.09 AUD) (Figure 2B). 

We also determined TRPV1 gene expression in the TG. The results showed that TRPV1 mRNA relative levels are higher in the TGs isolated from females in the diestrus phase than the proestrus phase and male mice (male: 1.0 ± 0.2, proestrus: 0.99 ± 0.09, and diestrus: 2.0 ± 0.13 relative levels) (Figure 3A). The analysis of TRPV1 protein expression revealed higher TRPV1 protein levels during the proestrus phase compared to the diestrus. The results also showed that male mice exhibit lower levels of TRPV1 protein than females in proestrus (males: 0.6 ± 0.096, proestrus: 0.87 ± 0.07, and diestrus: 0.5 ± 0.06 ADU) (Figure 3B).

### 2.3. Effects of 17β-Estradiol on Capsaicin Response on Trigeminal Neurons 

In addition, we evaluated the capsaicin-induced Ca^2+^ influx in primary TG neuron cultures treated overnight with low and high concentrations of 17β-estradiol (100 (LE2) and 200 pM (HE2), respectively), which resemble the steroid plasmatic concentration during the diestrus and proestrus phases, respectively [25]. 

The results showed no differences in the percentage of capsaicin-responsive neurons, as shown in Figure 4A. Also, minimal differences were observed in the amplitude of the responses induced by capsaicin and a high potassium solution (60 mM) between experimental groups. Interestingly, we observed changes in the capsaicin response latency (calculated as the time to reach peak responses after starting capsaicin perfusion). Neurons treated with LE2 concentrations (simulating diestrus) showed delayed responses to capsaicin compared to HE2 concentrations (simulating proestrus) (Figure 4B,C). These results suggest that the speed of neuronal excitability could be regulated by E2, with a faster response observed when E2 levels are high, as seen in the proestrus phase.

### 2.4. ERα Protein Levels in the TGs Are Higher in the Proestrus Than in the Diestrus Phase

Finally, we determined the ERα protein levels in the TGs isolated from male and female mice synchronized in the proestrus and diestrus phases. The WB analysis demonstrated that the ERα protein levels were higher in the TGs from female mice during proestrus than in the diestrus phase (P = 0.75 ± 0.17 and D = 0.41 ± 0.14), as shown in Figure 5. We found similar protein levels in male (0.87 ± 0.13) and female mice in the proestrus phase and a trend to higher ERα levels in male compared to female mice in the diestrus phase. The TRPV1 and ERα protein levels analysis showed that both proteins are higher in female mice in the proestrus than the diestrus phase, underlying the putative positive ERα role of regulating TRPV1 expression.

## 3. Discussion

Sex steroids play pivotal roles in regulating the physiology of nociceptors, affecting several painful conditions and the sexual dimorphism of pain [19,20,26,27]. 

For example, testosterone has antinociceptive effects in some animal models of pain, pointing out that this steroid confers pain resistance in the male sex [28]. On the contrary, several reports show that estrogens produce susceptibility to several painful conditions in the female sex [29]. The sex steroid actions activate specific receptors, which modulate the expressions and functions of several pain molecular players, such as ion channels. In this context, we previously established that progesterone (P4), a significant female steroid hormone, plays a crucial role in modulating capsaicin-mediated pain [30]. 

Here, we extended our investigation to evaluate the capsaicin-evoked pain thresholds in male and female mice during the estrous cycle, where sex steroid hormones fluctuate. 

Our behavioral acute pain tests revealed that capsaicin injection induced a more pronounced nociceptive response in female mice during the proestrus phase than in male mice. These findings are consistent with previous studies indicating that female rats exhibit heightened pain-related behaviors following capsaicin injection [20]. Similarly, this has been observed in humans, albeit evaluated through physiological responses rather than behavioral tests, resulting in women being more capsaicin responsive than men [19]. Thus, our results strengthen the presence of sexual dimorphism in capsaicin-induced pain, with variations dependent on the phase of the female estrous cycle. Some reports show that differences in pain thresholds between males and females may diminish under anxiety-free conditions [31]; however, substantial evidence indicates that mice undergoing estrogen treatments or cycling through high-estrogenic estrous phases (such as the proestrus) exhibit behavior indicative of heightened pain perception [22,32]. Our findings support the latter hypothesis, underscoring the significance of sex steroid hormonal contributions to pain perception at the physiological level.

We also demonstrated that capsaicin-induced pain threshold is lower during the proestrus than the other estrous cycle phases. This result is consistent with previous observations in rats, where capsaicin-evoked pain is higher during proestrus than estrus phases [20]. Furthermore, some experiments conducted in mice using different noxious stimuli, such as mechanical and heat inputs, have shown that female mice exhibit a higher nocifensive response during the proestrus than the estrus phase [21]. Interestingly, the ovariectomized female mice displayed a higher nocifensive threshold than proestrus female mice [21], suggesting that estrogens, the sex steroids prevalent in the proestrus phase, have a pivotal role in modulating the pain threshold. 

Estrogen’s positive actions on the regulation of capsaicin-induced pain have also been demonstrated in ovariectomized (OVX) rats with estrogenic replacement. These rats restore their capsaicin pain behavior compared to the lesser capsaicin effect in OVX rats [20,22,33].

Many research groups have been interested in elucidating the precise mechanism by which estrogens regulate capsaicin pain perception, whether through the modulation of TRPV1 function or expression. To elucidate this, we conducted quantitative PCR (qPCR) and Western Blot analyses to assess changes in TRPV1 expression during the estrous cycle when estrogen levels are high and low: the proestrus and diestrus phases, respectively, as well as in male mice. Our findings revealed similar TRPV1 mRNA levels in the DRGs of both male and female mice. TRPV1 mRNA expression was notably lower in the TGs of male and proestrus female mice than in the diestrus phase. These findings agree with a previous report, which also evaluated TRPV1 mRNA levels in different brain regions from female and male mice. This report identified the diestrus phase as having higher TRPV1 mRNA levels than the other estrous cycle phases [24]. Our results may appear contradictory to previous reports indicating the upregulation of TRPV1 mRNA in the TGs of OVX rats following high-dose E2 supplementation compared to low-dose E2 administration [22], since in our biological system, the proestrus and diestrus phases correspond to high and low doses of E2, respectively. A plausible explanation for the discrepancy between our results and those reported in the literature might be using ovariectomized rodents with replacement E2 regimens, indicating that the physiological role of E2 and other sexual steroids on TRPV1 mRNA expression could implied. The fact that the putative Estrogen Response Element (ERE) is in the promoter region of the mouse *Trpv1* gene [24] suggests that the estrogen receptor (α or β) could interact with this regulatory sequence to control the transcription of the TRPV1 gene. However, this hypothesis has not been probed.

We also evaluated the TRPV1 protein levels in TGs and DRGs in male, proestrus, and diestrus female mice. Our results show higher TRPV1 protein levels during proestrus than diestrus. This result correlates with the heightened paw-licking duration and wipe-bout count following capsaicin injection during the proestrus phase. These observations demonstrate that mRNA and protein expression do not always correlate due to post-transcriptional regulatory mechanisms [34]. We also found that the ERα protein levels are higher during the proestrus than the diestrus phase, suggesting an upregulation of both protein expressions when estrogen levels are high. ERα upregulation during proestrus has also been demonstrated in rat DRG neurons, confirming the dynamic changes in the expression of ERα and its putative target gene during the estrous cycle [35]. 

The data above strengthen the putative classic genomic estrogen actions to modulate TRPV1 gene expression, having as physiological consequence of susceptibility to pain response associated with this ion channel. The alternative way to estrogen’s actions could modulate the TRPV1 function, and some reports have suggested that E2 enhances capsaicin currents in DRG neurons [36]. We evaluated the capsaicin-evoked calcium influx on mouse TG primary cultures treated with low and high E2 overnight. We found that neither low nor high doses of E2 affected the calcium influx into cells. Interestingly, the capsaicin response velocity increased proportionally to the estradiol dose applied. This finding is consistent with reports indicating that E2 does not increase the total calcium influx in TRPV1-expressing cells after capsaicin-induced activation [37]. Nonetheless, these authors also reported that E2 decreased the threshold necessary for activation by capsaicin, which aligns with our observation regarding the increased speed of calcium influx.

This study has focused on studying the effect of sex steroid fluctuation on pain associated with TRPV1 activation, a crucial molecular target of several chronic painful conditions. Among the limitation of this study is the use of an acute pain model, which mainly evaluates nociception. However, this study opens a field in pain research to establish whether the same effects occur in chronic pain models where TRPV1 overactivation or effects over other nociceptive ion channels occur.

These findings provide insights into how sex steroid hormonal fluctuations modulate the pain associated with TRPV1 and contributes to the differential perception of capsaicin-induced pain between males and females. This difference underlies the symptoms of painful conditions such as orofacial pain, osteoarthritis, and migraines, which are more prevalent among females [13]. Although this study reports the influence of sex steroid hormonal fluctuations on acute pain produced by the activation of DRG and TG neurons, further research is needed to elucidate the molecular mechanisms underlying this regulation.

## 4. Materials and Methods

### 4.1. Animals

Male and female 10- to 12-week-old C57BL/6 mice were acquired from our institution’s animal facility. The procedures used in this project were approved by the Internal Committee for the Use and Care of Laboratory Animals (CICUAL, protocol SLM187-22) from the Institute of Cellular Physiology, Universidad Nacional Autónoma de México. Mice were maintained in a controlled environment with a 12 h light/dark cycle and access to food and water ad libitum. 

### 4.2. Determination of the Mouse Estrous Cycle

The estrous cycle phases were determined as previously reported [38]. Vaginal cellular samples from female C57BL/6 mice were collected through lavages with 20 μL saline filled tip. The sample was placed on glass slide and completely dried at room temperature. The smears were immediately stained with crystal violet (0.1%). Smear inspections were performed immediately after staining under a light microscope to determine the cell types present in each sample. 

### 4.3. Pain Behavior Assays

The mice were placed in individual plastic containers one hour before the experiment to adapt to the experimental conditions. A stock of capsaicin (Sigma-Aldrich, St. Louis, MO, USA) 20 μg/μL in ethanol was prepared; then, 0.28 μg/μL capsaicin (working solution) and 1.4% ethanol in saline (control solution) were prepared. Paw licking was evoked by intradermal capsaicin injection (2.8 μg) using a 33 G × 6 mm gauge needle. The animal in the control group were injected with 10 μL of 1.4% ethanol solution. The mice were observed for 10 min, and the cumulative licking time was recorded and reported as the paw licking time (PLT, sec). The capsaicin cheek injection was carried out using a 0.3 mL insulin syringe with a 33 G × 6 mm gauge needle. Experimental groups received either 0.1 μg capsaicin or 0.05% ethanol in saline (control) in a volume of 10 μL. The number of times that mice wiped their cheeks (pain behavior) was scored over 20 min.

### 4.4. DRG and TG Dissection

The animals were sacrificed by cervical dislocation and decapitated. Briefly, the animals were held in a dissection area. The animal body was collocated in the dorsal position for DRG dissection, the skin was retired, and the vertebral column was exposed and cut for DRG isolation. The ganglia were collected in phosphate-buffered saline (PBS) and placed on ice. For TG isolation, the skull was exposed to remove the brain. TGs located to the left and right were cut and placed in ice-cold PBS. After washing twice with cold PBS, the ganglia were homogenized in TRIzol (Thermo Fisher Scientific, Waltham, MA, USA) or lysis buffer.

### 4.5. TG Primary Cell Culture

After TG dissection, the ganglia were collected in ice-cold DMEM and washed twice. Ganglia were suspended in a solution containing 4 mg/mL collagenase type II and 1.25 mg/mL trypsin (Sigma-Aldrich, St. Louis, MO, USA) in DMEM ((Thermo Fisher Scientific, Waltham, MA, USA) and incubated for 10 min in a 37 °C water bath. The cell suspension was then centrifuged for 5 min at 200× *g,* and the supernatant was removed. Finally, the cell pellet was homogenized in 2 mL of prewarmed culture medium (DMEM with 10% Fetal Bovine Serum and penicillin/streptomycin, (Thermo Fisher Scientific, Waltham, MA, USA)) and seeded on poly D-lysine 1 mg/mL, (Sigma-Aldrich, St. Louis, MO, USA) coverslips. The cells were cultured at 37 °C in an atmosphere containing 5% CO_2_. Three hours after plating, the cells were treated with 100 or 200 pM of E2 (Sigma-Aldrich, St. Louis, MO, USA) overnight. 

### 4.6. Calcium Influx Measure

Intracellular calcium imaging in primary TG cell cultures was performed using the fluorescent indicator Fluo-4. Before each experiment, the cells were incubated with 4 μM Fluo-4-acetoxymethylester (Thermo Fisher Scientific, Waltham, MA, USA) for 60 min at 37 °C. Fluorescence measurements were made on a Nikon Eclipse Ti-U (Minato, Tokio, Japan) upright microscope fitted with an ORCA-Flash4.0 V3 camera (Hamamatsu, Shizuoka, Japan). Fluo-4 was excited at 488 nm (excitation time 100 ms) with the Lambda HPX-L5 LED light source (Sutter Instruments, Novato, CA, USA), and the emitted fluorescence was band passed at 510 nm. The images were acquired at 0.5 Hz frequency using HCImage software (https://hcimage.com/) (Hamamatsu, Shizuoka, Japan).

The increase in [Ca^2+^]_i_ was induced by adding capsaicin (500 nM). Neurons were also stimulated through local depolarization with a high K^+^ solution (in mM: KCl, 60; NaCl, 85; CaCl_2_, 1.8; MgCl_2_, 1.2; HEPES, 10; and glucose 10; pH 7.4 adjusted with NaOH; 300 mOsm) (Sigma-Aldrich, St. Louis, MO, USA). 

Image stacks were converted to TIFF and imported into FIJI 2.7.0. software (NIH, Bethesda, MD, USA) for post hoc analysis [39]. The amplitude of the responses elicited by capsaicin and the high K^+^ solution was expressed as ΔF = Fi − F0, where Fi represents the fluorescence at any given time, and F0 represents the mean basal fluorescence obtained during the first 10 s of recording. 

### 4.7. Western Blot Assays

DRGs or TGs were suspended and disaggregated in lysis buffer [150 mM NaCl, 10 mM Tris⋅HCl (pH 7.5), 1% TritonX-100] supplemented with 10 mM NaF and 1X Complete Protease Inhibitor (Roche). Lysates were centrifuged for 5 min at 1500× *g*, and supernatants were recovered for total protein quantification using a bicinchoninic acid assay (Sigma-Aldrich, St. Louis, MO, USA). Total protein extracts were separated using SDS-PAGE and transferred to PVDF membranes (Immobilon-P; Merck Millipore, Burlington, MA, USA) using a Hoefer Semi-Dry transfer system. Membranes were blocked with 6% nonfat dry milk in PBS and Tween-20 0.1% (PBS-T) and incubated overnight with goat polyclonal anti-TRPV1 (P-19, sc12498; Santa Cruz Biotechnology, Dallas, TX, USA) diluted 1/500 in PBS-T with 3% nonfat dry milk. For ERα analysis, a rabbit polyclonal antibody (ab75635, abcam) was diluted 1:500 in PBS-T with 3% nonfat dry milk and incubated overnight. A monoclonal antibody against GAPDH (14C10; Cell Signaling Technology, Danvers, MA, USA) diluted 1:5000 in PBS-T was used as the protein loading control. Proteins were visualized using immobilized antigens conjugated to horseradish peroxidase-labeled secondary antibodies and were detected using chemiluminescent substrate (ECL; Amersham Pharmacia, Amersham, Buckinghamshire, UK). Densitometric analysis was performed to determine the band intensity corresponding to TRPV1, ERα, or GAPDH signals. Thus, the graphed results represent the normalized TRPV1/GAPDH or ERα/GAPDH signals of the relative intensity of the detected and reported signals as arbitrary density units (ADUs).

### 4.8. qPCR

The expressions of TRPV1 and the GAPDH (endogenous control) genes were measured using real-time PCR. Briefly, total RNA was isolated from the TG or DRG using TRIzol reagent (Invitrogen, Carlsbad, CA, USA) following the supplier’s protocol. Then, 0.5 μg of total RNA was reverse-transcribed using random hexanucleotides and M-MLV reverse transcriptase (Invitrogen, Carlsbad, CA, USA), according to the manufacturer’s instructions. 

Real-time PCR was performed with the StepOne™ Real-Time PCR System using the Maxima SYBR Green/ROX qPCR Master Mix (2X) (Thermo Fisher Scientific, Waltham, MA, U.S.A.). The sets of primers used were as follows: TRPV1-F: 5′-AGGCCAAGACCCCAATCTTC-3′ and TRPV1-R: 5′-CAAACTCCACCCCACACTGA-3′; GAPDH-F: 5′-CTGTTGCTGTAGCCGTATTC-3′ and GAPDH-R: 5′-CTTGGGCTACACTGAGGACC-3′. The PCR thermocycling conditions were as follows: 60 cycles that included the following steps: 15 s of denaturation at 95 °C, 30 s annealing phase at 55 °C, and 30 s template-dependent elongation phase at 72 °C. The amplification of each sample was performed in at least three experiments with two technical replicates in the same PCR run. The differential gene expression of TRPV1 was calculated as the ratio normalized to the expression of the GAPDH gene. The data were analyzed using the equation to calculate the target amount = 2^−ΔΔCT^ [40]. To adjust for error due to biological variability in the data, we performed the analysis described by Willems et al. [41].

### 4.9. Statistical Analysis

Statistical comparisons were made using the one-way ANOVA and Tukey’s post hoc test (GraphPad Prism 7.0) for Figure 1, Figure 2, Figure 3 and Figure 4. *p* ≤ 0.05 was considered statistically significant. A two-way ANOVA with Šídák’s multiple comparison test was used in the statistical analysis of Figure 5. *p* ≤ 0.05 was considered statistically significant. Group data are reported as the mean ± SEM.

## Figures and Tables

**Figure 1 ijms-25-08040-f001:**
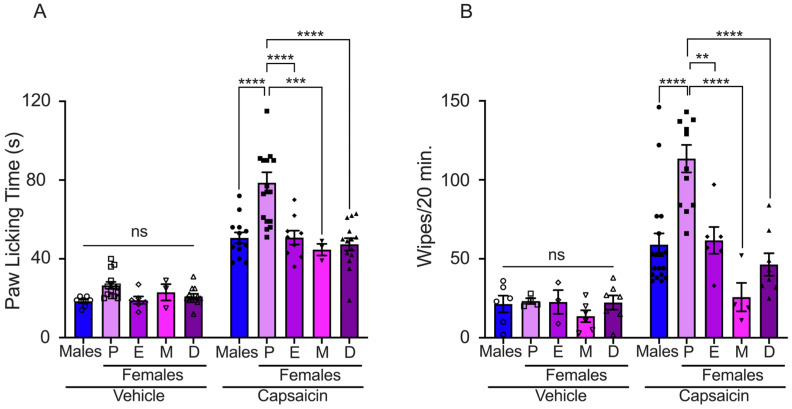
Evaluation of capsaicin-induced acute pain in male and female mice. (**A**) Paw licking assay: vehicle solution injection (bars with empty symbols) induced minimal paw licking response in all experimental groups (males: *n* = 6, P: *n* = 14, E: *n* = 6, M: *n* = 3, and D: *n* = 12). The nociceptive response was observed following capsaicin injection in all experimental groups (bars with filled symbols, males: *n* = 11, P: *n* = 17, E: *n* = 9, M: *n* = 3, and D: *n* = 14). (**B**) Cheek injection assay: The graph presents data generated from counting the number of wiping bouts following injection of either capsaicin or vehicle solution. Nociceptive response to capsaicin application was observed in all experimental groups (bars with filled symbols, males: *n* = 18, P: *n* = 12, E: *n* = 6, M: *n* = 4, and D: *n* = 8). The vehicle induced minimal wiping bouts in all experimental groups (bars with empty symbols, males: *n* = 6, P: *n* = 4, E: *n* = 3, M: *n* = 6, and D: *n* = 7). Proestrus (P), estrus (E), metestrus (M), and diestrus (D). Data are the mean ± SEM. One-way ANOVA followed by Tukey’s post hoc test, **, ***, and **** for *p* < 0.05, *p* < 0.001, and *p* < 0.0001, respectively; ns: non-statistically significant.

**Figure 2 ijms-25-08040-f002:**
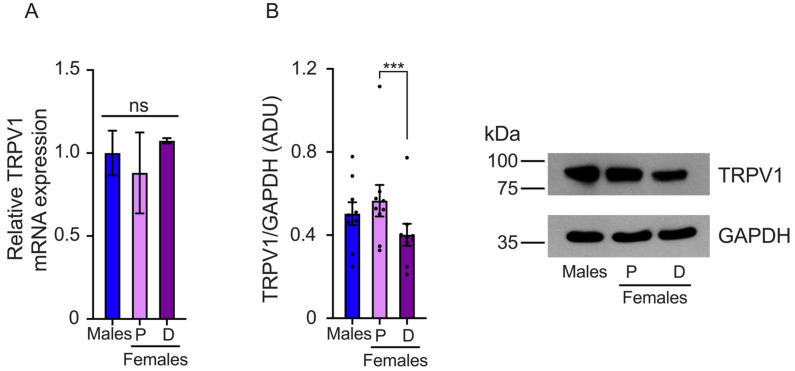
Evaluation of TRPV1 gene expression in the dorsal root ganglia (DRGs) of male and female mice in the proestrus (P) and diestrus (D) phases. (**A**) TRPV1 mRNA levels were assessed using qPCR. The data obtained showed similar TRPV1 mRNA levels across the three experimental groups. (**B**) Western Blot experiments demonstrated that TRPV1 protein levels were lower in the diestrus phase than in the proestrus phase (*n* = 9). Data are the mean ± SEM. One-way ANOVA followed by Tukey’s post hoc test. *** *p* < 0.005; ns: non-statically significant. ADU: arbitrary densitometric unit.

**Figure 3 ijms-25-08040-f003:**
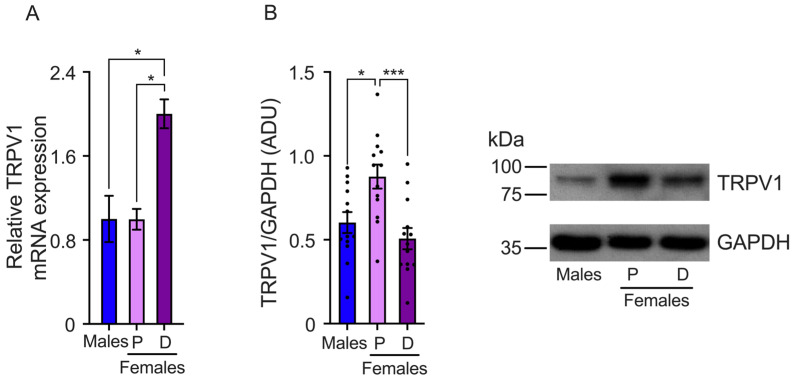
Evaluation of TRPV1 gene expression in the trigeminal ganglia (TGs) of male and female mice in the proestrus (P) and diestrus phases (D). (**A**) TRPV1 mRNA levels were assessed using qPCR. The data showed that TRPV1 mRNA levels are higher in diestrus than in proestrus and male mice (*n* = 3). (**B**) Western Blot experiments demonstrated that TRPV1 protein levels are lower in the diestrus phase compared to females in the proestrus phase (*n* = 13). Data are the mean ± SEM. One-way ANOVA test followed by Tukey’s post hoc test. * *p* < 0.05; *** *p* < 0.005. ADU: arbitrary densitometric unit.

**Figure 4 ijms-25-08040-f004:**
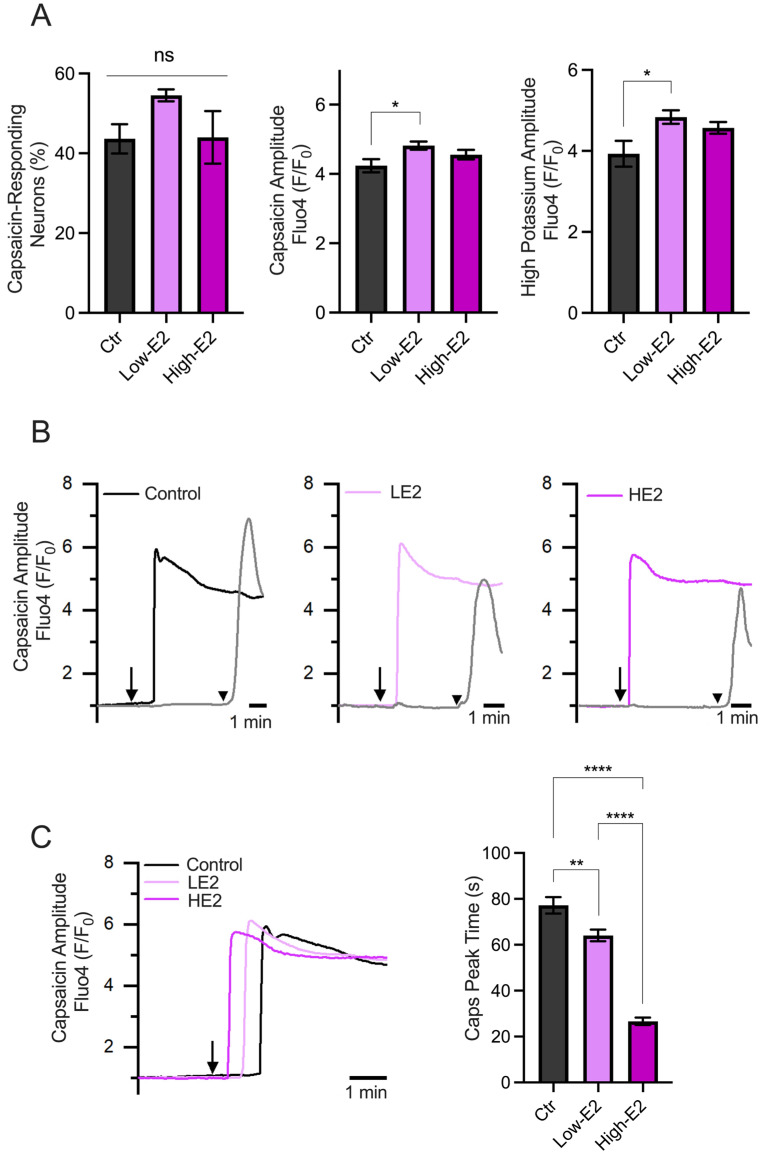
Calcium imaging recordings of TG neurons treated with low and high 17β-estradiol concentrations. (**A**) Summary graphs of the calcium imaging recordings performed on TG neurons in response to 500 nM capsaicin (left and middle) and high potassium (right) stimuli. The left graph shows the percentage of capsaicin-responding neurons under different experimental conditions: control condition (Ctr, *n* = 86), low E2 concentration (LE2, *n* = 125), and high E2 concentration (HE2, *n* = 117). The middle graph in A shows the amplitude responses to capsaicin, indicating that neurons from all groups respond similarly. The right graph in A displays the amplitude of the responses to the high potassium stimulus. (**B**) Representative recording of TG neuron groups, the capsaicin-positive neurons treated with HE2 show faster responses to capsaicin compared to those treated with LE2. The arrows represent the time capsaicin is added, and the arrowhead is when the high K^+^ solution is added. (**C**) Representative recording of the capsaicin response latency under each treatment (records taken from B) and the summary graph of the time to peak of the capsaicin responses. The bar graph represents the mean ± SEM. One-way ANOVA followed by Tukey’s post hoc test. *, **, and **** for *p* < 0.05, *p* < 0.01, and *p* < 0.005, respectively.

**Figure 5 ijms-25-08040-f005:**
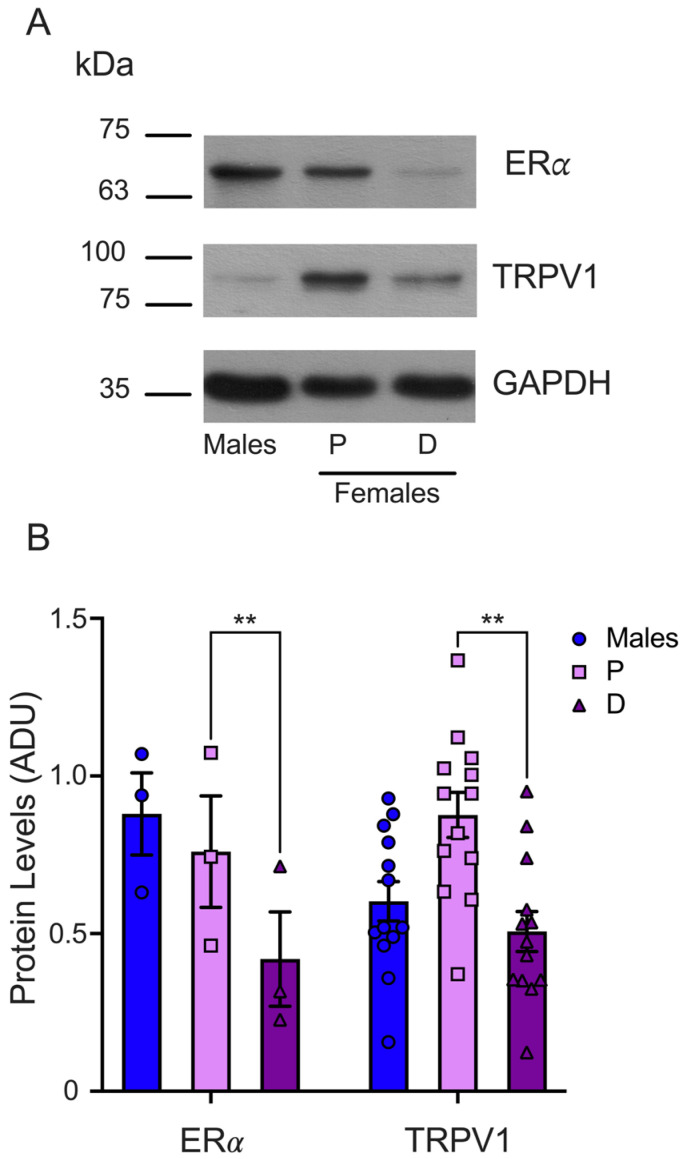
Evaluation of ERα total protein in TGs from male and female mice in the proestrus and diestrus phases. (**A**) Representative WB for ERα (upper panel) and GAPDH (lower panel) immunodetection. The membrane was stripped for TRPV1 immunodetection (middle panel). (**B**) The graph shows the normalized data of the mean values for ERα (*n* = 3) or TRPV1 (*n* = 13; these data are the same as those used in Figure 3) with respect to load control (GAPDH). Two-way ANOVA with Šídák’s multiple comparison test. ** *p* < 0.005. ADU: arbitrary densitometric unit.

## Data Availability

The current study’s data is available for request from the corresponding author.

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
