# Peer review of "The Influence of Sex Steroid Hormone Fluctuations on Capsaicin-Induced Pain and TRPV1 Expression"

_ijms, 2024, doi:10.3390/ijms25158040_

Round 1

Reviewer 1 Report

Comments and Suggestions for Authors

There is a growing interest in understanding how sex differences affect pain perception. It appears that differences in pain thresholds between the sexes are strongly influenced by sex steroid hormones, and TRPV1 is known to play an important role in pain perception.

In this study, the authors used two acute pain models in male and female mice during different estrous cycle phases to assess capsaicin-induced nociceptive responses and to evaluate TRVP1 gene expression. The authors find that there is a sexual dimorphism in capsaicin-induced pain in the two pain models. They also found that this difference is dependent on the estrous cycle phase of the female.

The authors report research that is highly informative about the modulation of pain by the female hormone estrogen. The paper provides very interesting results from both a medical and basic science perspective and is an interesting topic for publication in this journal. For the most part, the results are clear and the interpretation of the data seems reasonable.

Although the results are interesting, there is no analysis of the estrogen receptor, which is the receptor for estrogen, even though we are looking at the effects of estrogen; I think you should analyze the expression of estrogen receptors (alpha and beta) along with the expression of TRPV1. This would clarify which type of estrogen receptor is important and would greatly strengthen this paper.

Author Response

Answer to Reviewer 1

Although the results are interesting, there is no analysis of the estrogen receptor, which is the receptor for estrogen, even though we are looking at the effects of estrogen; I think you should analyze the expression of estrogen receptors (alpha and beta) along with the expression of TRPV1. This would clarify which type of estrogen receptor is important and would greatly strengthen this paper.

We thank the reviewer for their valuable comments and suggestions for our work.

We attended the suggestion about the analysis expression of some ER receptors (now Figure 5).  The expression of ERα and ERβ has been demonstrated in neurons from the TG and DRG (1-2). Thus, we evaluated ERα by WB finding that ERα total proteins are downregulated during the diestrus phase, like the TRPV1 protein.

This analysis is a now as a new figure (figure 5) and the description of these results are in lanes 215-222.

ERβ evaluation was not performed because the antibody used did not detect a specific signal in our extracts (see image in attached file ); thus, it remains to be determined in future experiments.

 We hope that these new data are according to reviewer suggestion.

Reference:

  1. Warfvinge K, Krause DN, Maddahi A, Edvinsson JCA, Edvinsson L, Haanes KA. Estrogen receptors α, β and GPER in the CNS and trigeminal system - molecular and functional aspects. J Headache Pain. 2020 Nov 10;21(1):131. doi: 10.1186/s10194-020-01197-0. PMID: 33167864; PMCID: PMC7653779.
  2. Taleghany N, Sarajari S, DonCarlos LL, Gollapudi L, Oblinger MM. Differential expression of estrogen receptor alpha and beta in rat dorsal root ganglion neurons. J Neurosci Res. 1999 Sep 1;57(5):603-15. PMID: 10462685.

Reviewer 2 Report

Comments and Suggestions for Authors

The introduction contains all the essential issues that should be explained to the readers, but still some points need improvement. In several cases, it is necessary to combine paragraphs for a better reading of the text. The description of the figures (especially Figure 1) needs to be improved and shortened. The discussion should be improved, taking into account the analyses performed and the conclusions drawn from them. In its current form, the discussion needs to be more precise in order to fully understand the results presented. It should also indicate which information in the literature is consistent and which is inconsistent with the results obtained. Also, the discussion section needs to include a section discussing the limitations of this study. The article contains minor language errors. English editing is suggested.

Comments on the Quality of English Language

Minor editing of English language required

Author Response

Answer to Reviewer 2

We appreciate the observations and suggestions of this reviewer. Below are the responses to each point:

  1. The description of the figures (especially Figure 1) needs to be improved and shortened.

We thank the reviewer for the valuable suggestions for our work. We have improved the description of all figures and combined some paragraphs for a better understanding.

  1. The discussion should be improved, taking into account the analyses performed and the conclusions drawn from them. In its current form, the discussion needs to be more precise in order to fully understand the results presented. It should also indicate which information in the literature is consistent and which is inconsistent with the results obtained.

We appreciate this observation. In the corrected version of this article, we have modified the discussion, comparing the data obtained in our study with previously published data in order to strengthen the contribution of this work as well as the conclusions drawn.

  1. Also, the discussion section needs to include a section discussing the limitations of this study.

Thank. We also added a paragraph at the end of the discussion mentioning the limitations of our work

  1. The article contains minor language errors. English editing is suggested.

Thank you for this suggestion. We have corrected the language errors.

  1. Comments on the Quality of English Language

Minor editing of English language required

We have conducted a careful review and correction of the English grammar.

We hope that our revised and corrected manuscript version attend all your valuable suggestions.

Reviewer 3 Report

Comments and Suggestions for Authors

Below are my comments and suggestions:

Weaknesses/Suggestions for improvement:

  1. The sample sizes for some experiments seem small (e.g. n=3 for qPCR). Larger sample sizes would increase confidence in the results.
  2. Statistical analyses could be more robust. The authors should consider using two-way ANOVA to analyze the effects of both sex and estrous cycle phase simultaneously.
  3. The discussion section could be expanded to more thoroughly interpret the results in light of existing literature and propose potential mechanisms for the observed effects.
  4. Some figures lack clear labeling and could be improved for clarity (e.g. Figure 4 is complex and difficult to interpret).
  5. The authors should address potential limitations of their study, such as using only one pain model (capsaicin) and not exploring other sex hormones beyond estradiol.
  6. The manuscript would benefit from careful proofreading to correct minor grammatical errors.
  7. The conclusion could be strengthened by more explicitly stating the implications of this work for understanding sex differences in pain and potential clinical applications.

Overall, this is a solid study that provides valuable new data on an important topic. With some refinements to the analysis and presentation, it could make a strong contribution to the field. 

Author Response

We thank this reviewer for careful assessment of our manuscript. We hope that we have addressed all the reviewer’s concerns. Please see below.

  1. The sample sizes for some experiments seem small (e.g. n=3 for qPCR). Larger sample sizes would increase confidence in the results.

We appreciate and understand the reviewer's concern. We performed quantification experiments according to the article Real-Time Quantitative RT-PCR: Design, Calculations, and Statistics (ref 1). The authors mention that to perform an experiment, at least three independent biological replicates of each condition should be included, and for each biological replicate, it is usual to perform at least two technical replicates of each reaction. The main problem with this type of experiment is the biological variation between each sample. To corroborate the results obtained by RT-qPCR, we performed a new analysis using a method that is used when there may be large variability in the data (ref. 2). Interestingly, the results obtained in this analysis were the same as those we reported, suggesting that the observed differences were correct and reliable. Section 4.8 qPCR adds text and citations for this analysis.

  1. Statistical analyses could be more robust. The authors should consider using two-way ANOVA to analyze the effects of both sex and estrous cycle phase simultaneously.

We thank the Reviewer for the comment.

We use one-way analysis of variance (ANOVA) followed by Tukey´s post hoc tests since we are considered one dependent variable (the pain threshold) and five comparative groups (males, proestrus, estrous, metestrus and diestrus). The Tukey pos hoc test allows us to detect statistical differences between all the groups, even the difference between males and females in different phases of estrous cycle. Our results clearly show changes in capsaicin pain threshold during estrous cycle making difficult to compare only male and all females.

The use of a two-way ANOVA is not feasible in our experimental design used for figure 1 because we only considered one factor. Although males do not strictly belong to any phase of the estrous cycle, for the sake of statistical analysis, we considered them as an additional group within this factor, a group with a minimal contribution of hormonal fluctuation to the system and, therefore, to the evaluated behavior. A two-way ANOVA could be viable if we had a second factor, but this is not possible in our case with the factor “sex” because there would be only one group in “males” of this second factor, making difficult a viable comparison (references 3,4,5).

For the statistical analysis of the new results added in figure 5 we used a two-way ANOVA in which we compare the protein expression levels of TRPV1 and ER? (one factor for the ANOVA) in male mice, and female mice in the proestrus and diestrus phases (the second factor for the ANOVA) hence making this analysis adequate for the data we presented in this new figure.

  1. The discussion section could be expanded to more thoroughly interpret the results in light of existing literature and propose potential mechanisms for the observed effects.

Thank you for the comment. We have modified the discussion section considering more literature to propose a putative mechanism.

  1. Some figures lack clear labeling and could be improved for clarity (e.g. Figure 4 is complex and difficult to interpret).

Thank for the comment. We have modified the labels of the figures 1-3 (minor changes) and we have changed all the labels of figure 4 to better clarity for the reader.

  1. The authors should address potential limitations of their study, such as using only one pain model (capsaicin) and not exploring other sex hormones beyond estradiol.

We agree. Now, we add a paragraph in the discussion section about the limitation of our study.

  1. The manuscript would benefit from careful proofreading to correct minor grammatical errors.

Thank you for the suggestion. We agree with this comment. We have made careful revision to correct the grammatical errors.

  1. The conclusion could be strengthened by more explicitly stating the implications of this work for understanding sex differences in pain and potential clinical applications.

We Agree. We have done a more explicitly and strengthened conclusion of our work and its relevance in pain application.

  1. Overall, this is a solid study that provides valuable new data on an important topic. With some refinements to the analysis and presentation, it could make a strong contribution to the field

 We thank to reviewer for your constructive feedback.

References

  1. Rieu I, Powers SJ. Real-time quantitative RT-PCR: design, calculations, and statistics. Plant Cell. 2009 Apr;21(4):1031-3. doi: 10.1105/tpc.109.066001. Epub 2009 Apr 24. PMID: 19395682; PMCID: PMC2685626.
  2. Willems E, Leyns L, Vandesompele J. Standardization of real-time PCR gene expression data from independent biological replicates. Anal Biochem. 2008 Aug 1;379(1):127-9. doi: 10.1016/j.ab.2008.04.036. Epub 2008 Apr 26. PMID: 18485881.
  3. Pandis N. Analysis of variance. Am J Orthod Dentofacial Orthop. 2015

Nov;148(5):868-9. doi: 10.1016/j.ajodo.2015.08.009. PMID: 26522048.

  1. Pandis N. Two-way analysis of variance: Part 1. Am J Orthod Dentofacial Orthop. 2015 Dec;148(6):1078-9. doi: 10.1016/j.ajodo.2015.09.015. PMID: 26672715.
  2. Pandis N. Two-way analysis of variance: Part 2. Am J Orthod Dentofacial Orthop. 2016 Jan;149(1):137-9. doi: 10.1016/j.ajodo.2015.10.007. PMID: 26718388.
